# Age-Specific Effects of Visual Feature Binding

**DOI:** 10.3390/brainsci13101389

**Published:** 2023-09-29

**Authors:** Michelle Werrmann, Michael Niedeggen

**Affiliations:** Research Unit General Psychology and Neuropsychology, Department of Education and Psychology, Freie Universität Berlin, 14195 Berlin, Germany; m.werrmann@fu-berlin.de

**Keywords:** feature binding, visual working memory, aging, ERP

## Abstract

Temporary binding of visual features enables objects to be stored and maintained in the visual working memory as a singular structure, irrespective of its inherent complexity. Although working memory capacity is reduced in aging, previous behavioral studies suggest that binding is preserved. Using event-related brain potentials (ERPs), we tested whether stimulus encoding is different in younger (N = 26, mean age = 28.5) and older (N = 22; mean age = 67.4) participants in a change detection task. The processing costs of binding were defined by the difference between feature-alone (color or shape) and feature-binding (color–shape) conditions. The behavioral data revealed that discrimination ability was reduced in the feature-binding condition, and that this effect was more attenuated in older participants. A corresponding ERP effect was not found in early components related to visual feature detection and processing (posterior N1 and frontal P2). However, the late positive complex (LPC) was more often expressed in the feature-binding condition, and the increase in amplitude was more pronounced in older participants. The LPC can be related to attentional allocation processes which might support the maintenance of the more complex stimulus representation in the binding task. However, the selective neural overactivation in the encoding phase observed in older participants does not prevent swap errors in the subsequent retrieval phase.

## 1. Introduction

Visual objects are usually defined by different features such as color, shape, or position. To form and maintain an internal representation, the specific conjunction of object features must be stored in a bound form (feature binding [1]). The seminal work of Luck and Vogel [2] indicated that the units in visual working memory are defined by integrated objects rather than single features. However, the creation and maintenance of an integrated object is vulnerable to interference [3] and depends on the allocation of attentional resources [4].

The question of whether the binding process activates a distinct processing stage and/or neural substrates is still open (see [5]). Electrophysiological and neuroimaging findings focusing on the construction and maintenance of feature maps in the binding process are not conclusive. Following a previous event-related brain potentials (ERP) study [6], three components were attenuated in a feature-binding (shape-to-color) condition as compared to a feature-alone (shape or color) condition: first, the posterior N1 component is generated in the extrastriate cortex and can be modulated by attention [7]. In complex visual tasks, the N1 has been assumed to reflect the processing of a visual stimulus configuration [8]. Changes in the N1 amplitude have previously been related to the success of a working memory program [9]. Second, the fronto-central P2 component has been related to the perception and processing of salient stimuli, and its expression also depends on the complexity of visual stimuli [10,11]. In working memory tasks, P2 amplitude might serve as an index of the attentional resources used in the encoding procedure [12]. Finally, a late positive component (LPC) has been identified in a wide range of cognitive tasks. In memory tasks, the LPC is assumed to reflect the engagement of elaborative encoding and storage processes [13]. In working memory, the expression of the LPC is assumed to reflect the recruitment of attentional resources [14] and the maintenance of working memory [15]. In sum, these results can be related to a deficit in attentional control in the encoding phase. However, these effects rely on the examination of patients with mild cognitive impairments (MCI), which are defined as risk factors for a progredient neurodegenerative disease. Corresponding brain imaging studies in healthy participants emphasize that feature binding relies on the maintenance of visual information in the ventral visual system [16,17].

A further ongoing debate is related to the effect of aging on binding performance: although the capacity of working memory declines with advanced age [18], a number of studies suggest that feature binding reflects an independent process which is not affected by an age-specific decrement [19,20,21,22,23,24]. This contradicts the predictions of a two-component model of binding [4] incorporating a low-level feature-binding process (i.e., perceptual integration in the visual cortex) supported by a top-down control (i.e., feedback processes from higher cortical regions). Both processes are assumed to be affected by maturational changes, and corresponding evidence has been reported in experimental studies [25,26,27,28].

The rationale of our experimental approach relied on these ongoing debates: can we identify a distinct process associated with the binding of distinct visual features, and is the activation of this process modulated by aging? These research questions are also of clinical relevance: as pointed out by Logie and colleagues [29], the binding task is a sensitive neuropsychological marker in Alzheimer’s disease (AD). More importantly, the task supports the differentials diagnosis since selective binding deficits cannot be observed in aging or depression [30].

We used ERPs to identify binding-specific processes in the encoding and maintenance of color and shape information. In two groups of participants (younger vs. older), the internal representation of single (feature-alone) or combined (feature-binding) visual features was assessed by running a change detection task established in previous studies [22].

Our first research question addressed the electrophysiological effects of a binding process during stimulus encoding and/or maintenance. Previous neuroimaging findings supporting the role of low-level binding processes indicate a specific activation in the ventral visual system in a feature-binding condition [16,17]. Consequently, we hypothesized that binding costs affect the expression of early posterior components related to the processing of visual configurations, such as the N1 [6]. If binding processes also imply top-down control processes, as suggested by the two-component model [4], the late positive complex will be enhanced [6].

The second research question was related to the modulation of binding effects by the ‘aging’ factor. In line with the two-component model of binding, we expected that the costs of visual feature binding are higher in older individuals than in younger individuals [25,26,27,28]. The expected behavioral effects in discrimination performance should also reflect itself in the ERP data: the effect of aging on low-level binding processes will primarily affect early ERP components (N1, P2). Age-dependent differences in the elaboration of encoding will affect the expression of late ERP components (LPC), and these are likely to be affected. Corresponding compensatory brain processes in aging (see [31]) have previously been reported in older participants [32,33].

## 2. Materials and Methods

### 2.1. Participants

The experimental procedure was approved by the local ethics committee at FU Berlin (212/2018). No part of the study procedure was pre-registered. The sample size was determined a priori using G*Power [34]: based on previous results reporting age-specific effects in a visual working-memory task [35], we aimed to detect a medium effect (f = 0.20 adjusted to the taxonomy of Cohen) of the interaction of the within-factor ‘condition’ (feature-color vs. feature-shape vs. feature binding) and the between-factor ‘age’ (younger vs. older individuals) with a power of 80%. Using an F-test with alpha at 0.05, a sample size of 42 participants was required.

In total, the data of 59 participants were recorded. The data of participants reporting problems with vision (e.g., uncorrected myopia, color vision deficiency), neurological and/or psychiatric disorders, current psychoactive medication or drug dependence were excluded. At least one of these criteria applied to two participants. Following a rigorous artefact rejection in EEG analysis (criteria: see Section 2.3), the data of nine participants were not considered in the final analysis. Older participants additionally completed the German version of the Mini Mental State Examination MMSE (ref. [36]) to rule out cognitive impairments (score > 26/30 points. Test Scores of older participants: Appendix A).

The final sample comprised a total of 48 participants. The group of younger participants comprised 26 participants (19 female, 7 male, mean age: 23.85, SD: 4.38, range: 19–33 years), and the group of older participants 22 participants (14 female, 8 male, mean age: 68.77, SD: 4.99, range 60–77 years).

### 2.2. Task and Design

The experimental task (see Figure 1A) used the change detection paradigm established by Brockmole and colleagues [22] which includes two feature-specific (color, form) and a feature-binding condition. In each trial, the visual array included three stimuli presented at a randomly chosen spatial position on a computer screen (Sony, 21”, extension: 2.9° × 2.9° at a viewing distance of 100 cm). Stimuli were defined by abstract geometrical shapes in order to allow for a comparison with previous findings [21,22]. Timing of the stimulus sequence (programmed in PsychoPy, v1.8, [37]) followed the protocol defined by [38]. Each trial was defined by the presentation of the study array (2000 ms) followed by a blank screen (900 ms). The presentation of the subsequent test array was determined by the response of the participant. Following the participant’s response, a blank screen was presented for another 1000 ms.

In the feature-specific condition color, three stimuli of an identical shape were presented in different colors (randomly chosen out of a pool of eight different colors) in the study array. In the subsequent test array, the spatial position of the three stimuli was changed. In the match condition (probability: 50%), colors were identical with the study array. In the mismatch condition (probability: 50%), two colors were changed. Participants were required to indicate change detection (yes/no) by pressing a corresponding pre-defined button on a keyboard.

In the feature-specific condition shape, the three stimuli in the study array did not differ in color (black), but in shape (randomly chosen out of a pool of eight different shapes). In the subsequent test array, the stimuli were rearranged with respect to spatial position. In the match condition (probability: 50%), shapes were identical with the study array. In the mismatch condition, two shapes were changed (mismatch, 50%). Change detection was to be reported by participants as described above.

In the condition feature binding, the three stimuli in the study array differed in shape and color. Participants were instructed to maintain the intra-item assignment of each shape to its color. In the test array, the color–shape assignment of the spatially rearranged stimuli was either identical with the study array (match, 50%), or for two items, the color was swapped (mismatch, 50%). Please note that the test array therefore never included a shape or color not included in the preceding study array.

Each condition was tested in an experimental run defined by 32 trials, and it was preceded by a practice session (16 trials per condition). The order of conditions was quasi-randomized, but the binding condition was always presented last. Pilot studies revealed that discrimination performance in older participants approached the level of chance if feature binding was already required in the first run. To test for possible training effects, each participant underwent the experimental protocol twice (factor ‘block’). The repetition of the sequence allowed us to test the reliability of age-specific binding effects.

All participants also underwent a neuropsychological test battery focusing on visuospatial cognition (e.g., block span, Trail Making Test, visual memory). All data are available in the data repository.

The experimental setup—reduced to one run and excluding the EEG recording—was tested in an independent sample of older participants (N = 102 participants, aged between 50 and 87 years). Data from this pilot sample are also available in the data repository.

### 2.3. EEG Recording

The EEG was simultaneously recorded at five active electrode leads (Fz, Cz, Pz, T5, T6; impedance kept below 5 kOhm) referenced to linked earlobes. Ocular artefacts were controlled for by recording vEOG and hEOG. The online signal was continuously sampled at 500 Hz, and band-pass filtered (0.1–100 Hz).

The ‘Vision analyzer’ (version 2.1, Brain products, Gilching, Germany) was used for offline analysis, and was focused on the ERPs triggered in the study array, which can be linked to the encoding phase: the EEG signal was filtered (0.3 to 30 Hz, 12 dB/Oct), epoched according to the onset of the study array (−100 to 900 ms), and baseline-corrected (−100 to 0 ms). Artefact-free trials were averaged separately for the factors ‘condition’ (color, shape, and binding), ‘electrode position’, and ’block’ (1st and 2nd run). In the preceding artefact control, single trials were rejected in the case of movement artefacts, ocular artefacts (blinks, vertical or horizontal eye movements > 4°) or EEG alpha activity (>80 μV). The analysis only considered trials in which the participant responded correctly (hits, correct rejections). Participants were excluded from the analysis if the average relied on less than 12 trials in a block (9 out of 59 participants). The mean number of trials was 34.2 [=53.4%] (color: 36.8 [=57.5%], shape: 34.3 [=53.5%], binding: 31.4 [=49.1%]) in older, and 44.5 [=69.5%] (color: 49.5 [=77.3%], shape: 42.8 [=66.9%], binding: 41.3 [=64.5%]) in younger participants.

The ERPs triggered by the onset of the test array were also analyzed. As indicated by the grand-averaged ERPs presented in Appendix A, binding-specific effects cannot be observed. This also applies for the time range of the N400, in which binding-specific effects were reported [39].

### 2.4. Data Analysis

The statistical analysis of the behavioral data was based on the discrimination index A’, [40] considering the proportion of the probability of hits (H) and false alarms (F). In the case of H > F, A’ was estimated by:A’ = 0.5 + [(H − F) (1 + H − F)/4H (1 − F)]

In order to allow for a comparison with previous data [6], analysis of the ERP data was focused on three components: N1, P2 and LPC. In order to consider age differences in the latency of the more transient components (here: N1 and P2), temporal windows for the computation of mean activity were slightly different for older and younger participants (N1: 170 to 210 ms for older participants; 165 to 205 ms for younger participants; P2: 220 to 260 ms for older participants; and 215 to 255 ms for younger participants). The grand-averaged ERP (see Figure 2A) indicated that the N1 was expressed at temporal leads (T5, T6), and that the P2 was expressed at fronto-central leads (Fz, Cz). Please note that we additionally applied an analysis of peak amplitudes for both components (module ‘peak detection’ in ‘Vision analyzer’ version 2.1, Brain products, Gilching, Germany). Data are available in the data repository and confirm the results of the analysis of mean amplitudes reported in the following.

For the more sustained LPC, a single peak could not be determined in the majority of participants. The global field power (GFP) indicated a longstanding activity between 500 and 650 ms. Within these temporal regions, the mean activity was determined in each participant. The LPC was mostly expressed at centro-parietal sites (Cz, Pz).

Behavioral (A’) and ERP (N1, P2, LPC) data were analyzed running a mixed analysis of variance (ANOVA) using SPSS (IBM Statistics for Windows, Version 22, IBM Corp.), including the within-participant factors, ‘condition’ and ‘block’ and the between-participant factor ‘group’. In the case of a significant interaction, post hoc comparisons were computed: a binding-specific effect required that the binding condition was significantly different from both feature-alone conditions (color, shape).

In a final exploratory analysis, we analyzed whether the binding-specific effects observed in the ERPs (LPC amplitude) could be related to the behavioral discrimination performance (A’). To this end, the net effects of binding were computed for the ERP positivity (Δ_SB_ = LPC(shape) − LPC(binding), Δ_CB_ = LPC(color) − LPC(binding)) and behavioral variables (Δ_SB_ = A’(shape) − A’(binding), Δ_CB_ = A’(color) − A’(binding)). For the corresponding net effects (Δ_SB_ and Δ_CB_), correlation coefficients (Pearson) were determined and separated for the two groups.

## 3. Results

The descriptive results of behavioral and ERP data are depicted in Table 1 and in Figure 1B and Figure 2B. In accordance with our hypothesis, the results are focused on the main effect of ‘condition’ and on the crucial interactions between the factors ‘condition’ and ‘age’. If the superordinate ANOVA signaled a significant effect, the results of the post hoc comparisons are presented in Table 2. Please note that a main effect of ‘age’ was observed for each behavioral and ERP variable, indicating a general reduction in discrimination performance and amplitude. The main effects of the experimental factor ‘block’ were also obtained in the behavioral and ERP data. Importantly, the factor ‘block’ did not interact with the factor ‘condition’, nor with the interaction term ‘condition x age’. Given the reliability of the effects, behavioral and ERP data were collapsed for the two blocks. The extended results of the ANOVA—including the factor ‘block’—are provided in Appendix A.

### 3.1. Behavioral Data

As depicted in Figure 1B, hit rates can be misleading when estimating age-specific effects on performance in a change detection task: problems in feature binding were more prominent in false alarms than in hit rates, and this effect was more often expressed in older participants: As contrasted to the false alarm rate in the feature-specific condition ‘shape’, the mean rate increased from 5.7% to 17.8% in younger participants, and from 16.8% to 44.6% in older participants (for analysis of hits and false alarms: see Appendix A).

To account for this effect, the analysis therefore focused on the discrimination ability A’ [19]. Mean A’ was significantly reduced in feature binding when compared with the feature-specific conditions ‘color’ and ‘shape’ (see Table 1, Figure 1C). This effect was more often expressed in older than younger participants—as indicated by the significant interaction of the factors ‘condition’ and ‘age’ (see Table 2).

The post hoc analysis for older and younger participants (Table 2) revealed that a binding-specific effect was expressed in both groups. To determine whether the costs of feature binding were more often expressed in older participants, we determined the net effects (Δ_SB_ = A′(shape) − A′(binding), Δ_CB_ = A′(color) − A′(binding)) in each group. Both of the net effects of the bindings were larger in older (means: Δ_SB_ = 0.11, Δ_CB_ = 0.18) compared to younger (means: Δ_SB_ = 0.03, Δ_CB_ = 0.05) participants, and the differences were significantly expressed in both contrasts (Δ_SB_: F(1,46) = 13.29, *p* = 0.001, η_p_^2^ = 0.224; Δ_CB_: F(1,46) = 36.53, *p* < 0.001, η_p_^2^ = 0.443).

### 3.2. ERP Data

In line with the previous ERP study [6], the processing of the study array elicited a series of prominent components: the temporal N1, fronto-central P2, and the sustained centro-parietal LPC. The grand-averaged ERP data and the assignment of the components to temporal windows of analysis are presented in Figure 2.

Temporal N1 (170–210 ms): the descriptive analysis (Table 1, Figure 2) showed an increase in amplitude in the shape condition (shape > binding and shape > color). This effect was confirmed in the ANOVA and in the post hoc comparisons (Table 2). Crucially, no differences were observed between the binding and color conditions. Moreover, the main effect of condition was not modulated by the factor ‘age’. In sum, the N1 amplitude reflects a general increase when attention is directed to the shape of the stimuli. Please note that the result of the peak analysis (see Appendix A) confirms this result. An effect of N1 latency was not observed.

Fronto-central P2 (220–260 ms): the positive shift at the fronto-central leads was pronounced for the color condition (Table 1). Post hoc comparisons revealed a significant difference in the binding, but not in the shape condition. Importantly, no differences were observed between the binding and shape condition (Table 2). The significant interaction between the factors ‘condition’ and ‘age’ signaled that the significantly increased activation observed in the color condition was only expressed in younger participants. In older participants, no differences between the conditions were observed (Table 2). This indicates an age-specific effect in the encoding of color stimuli. This effect was confirmed by the peak analysis (see Appendix A). An effect of N1 latency was not observed.

Centro-parietal LPC (400–550 ms): the positive amplitude was significantly more pronounced in the binding compared to both feature-alone conditions (binding > color and binding > shape). The main effect of the factor ‘condition’ (Table 2) therefore indicated a binding-specific effect. Importantly, the expression of the effect of ‘condition’ was modulated by the factor ‘age’. Post hoc comparisons within each group revealed that the significant enhancement of the LPC amplitude in the binding condition indicated by the main effect was only significantly expressed in the group of older participants. In younger participants, activation in the binding condition did not differ significantly from the feature-specific processing of shapes.

Taken together, a binding-specific ERP activation separable from feature-specific processing (binding > color and binding > shape) was not observed in early components. The expression of the LPC was enhanced if feature binding was required, but this ERP effect was only significantly expressed in older participants.

### 3.3. Relation between ERP and Behavioral Data

In a final exploratory analysis, the costs for binding expressed in the LPC were related to the discriminative performance (A’). Costs of binding were estimated as net effects of the ERP positivity (Δ_SB_ = LPC(shape) − LPC(binding), Δ_CB_ = LPC(color) − LPC(binding)) and discrimination performances (Δ_SB_ = A’(shape) − A’(binding), Δ_CB_ = A’(color) − A’(binding)) were computed. When the costs of feature binding were contrasted to the feature-specific condition shape (Δ_SB_), correlations were not expressed in younger participants (Δ_SB_: r = 0.36, *p* = 0.065) nor were they expressed in older participants (Δ_SB_: r = 0.19, *p* = 0.404). The same pattern applied for the costs of binding compared to color processing (Δ_CB_): Correlations were not significant in the group of younger participants (Δ_CB_: r = 0.36, *p* = 0.065) nor were they significant in the group of older participants (Δ_CB_: r = −0.11, *p* = 0.635).

## 4. Discussion

Based on the results, our initial research questions can be answered as follows: in the ERP data, binding-specific effects (binding > color and binding > shape) were exclusively expressed in the late positive component (LPC), but not in earlier ERP components related to stimulus encoding. The binding-specific ERP effect was more often expressed in older as compared to younger participants. A corresponding group difference was also observed for the discrimination performance, which was more attenuated in older participants if feature binding was required. However, the ERP effect does not serve as a predictor for the behavioral performance. These findings will be discussed in the following section.

### 4.1. ERP Effects of the Binding Process

The change detection tasks used in this study replicated the previously reported costs of feature binding [16]. However, the expected modulation of early visual components (here: N1) was not observed. This contrasts previous findings [6], and questions the role of a low-level feature-integration process.

We suppose that differences in feature-specific processing must be considered: the effect in the N1/P2 range reported previously [6] was exclusively based on the contrast between the feature binding and one feature-specific (shape) condition. Please note that a corresponding ERP effect can be observed in our data when the feature-specific conditions are contrasted (shape vs. color, see Table 1 and Table 2). However, the ERP effect does not apply to the contrast of feature binding and the feature-specific ‘color’ condition. Differences in feature-specific conditions were also reported in a neuroimaging study on feature binding: differences in the encoding of shape and color were observed in the fusiform activity [16]. In sum, these findings emphasize that any interpretation of neural and behavioral feature-binding effects must also consider differences in feature-specific processing.

Our data suggest that binding processes based on intra-item association of color and shape do not enhance a neural process associated with stimulus encoding. Comparable results were reported in neuroimaging [41] and ERP studies: the expression of the CDA (contralateral delay activity) was related to the number of features to be maintained, but not to the conjunction of features [42].

Feature-binding-specific processes were restricted to an enhancement of the LPC amplitude. This supports the notion that bindings require engagement in elaborative encoding and storage processes [12]. This process is likely affected by maturational processes, as discussed in the next section.

### 4.2. Age-Related Modulations on the Binding Process

The ERP data allow us to approach the question of whether the sensory and cognitive processes underlying feature binding are modulated by aging. Early components (N1, P2) signal that the early stimulus processing of the feature ‘color’ can be specifically modulated by the factor ‘age’; however, selective effects related to feature binding were restricted to an enhancement of the LPC amplitude in older participants.

The age-specific ERP effect can be linked to selective neural overactivation, which might serve a compensation process [42]. The centro-parietal LPC has already been observed in working memory tasks and has been associated with resource allocation [13,14]. Greater frontal expression of this component has been related to the maintenance of working memory content [43]. Correspondingly, the increase in the LPC in the group of older participants might signal that participants recruit more attentional resources in the encoding of the intra-item association and/or that neural networks are more engaged in the attempt to maintain the internal representation of the complex stimuli [44] In contrast to the aging-related frontal ERP effects observed in working memory tasks [32,33], the posterior expression of the LPC effect suggests that the compensation process in feature binding is probably linked to a different neural network.

Importantly, this ERP effect is not correlated with the age-specific effects in feature binding. We also observed that performance in the binding task was significantly more attenuated in older, compared to younger, adults. As pointed out previously, this age-specific effect is not undisputed [21,22,24]. A summary of the previous results (see Appendix A) indicates that positive findings (i.e., evidence for an age-specific binding effect) do not depend on the properties of the sample (sample size, age), differences in design (e.g., presentation time), or on the order of conditions. Moreover, our data provide evidence that age-specific effects are not modulated by training: the more pronounced attenuation of discrimination performance was consistently observed in the two successive experimental blocks (see Appendix A).

Our data also indicate that, in older participants, the loss in discrimination ability in binding is primarily due to an increase in the false alarm rate (see Figure 1B), and therefore reflects a limited precision in recall. These recall errors might be due to maturational changes in the visual, hippocampal, and prefrontal network [4] which contribute to a noisy internal representation of features in older participants [5]. Since the test array in the feature-binding condition does not includes new shapes or colors, a less precise internal representation of the objects in the study array will promote a swap of features. This explanation is in line with the idea that increased false recognition rates in older people [44] are related to differences in novelty-processing in visual memory tasks [35].

Notably, this process is not directly related to the aforementioned LPC enhancement in older adults. The ERP effect was not correlated with discrimination performance in older, or in younger, participants. Therefore, it is questionable whether the increase in neural activity reflects an effective neural compensation process. In contrast to previous encoding-related ERP effects [45], the increase in LPC activity observed in older participants is not associated with an increase in discrimination performance. This is also highlighted by the posterior topography of the LPC effect: aging-related ERP effects associated with an increase in the performance in working memory tasks have been related to frontal processes [31,32]. This indicates that the LPC effect observed in the older participants in this study is related to a different neural network that is not engaged in an effective neural compensation [46].

### 4.3. Limitations

First, we must consider constraints on the generality of our findings related to the sample of participants. Older participants were overall interested in their cognitive performance, highly motivated, and the educational level in this group was rather high (mean 11.23 years of education). Therefore, it remains to be explored if the compensatory effort expressed in the ERPs can also be observed in a group of participants with a lower level of education.

Second, we must consider that discrimination ability, with respect to the shape condition, was already reduced in older participants. This effect might be due to the choice of abstract geometrical shapes used in previous binding studies to control for effects of verbal rehearsal [22]. However, the database from our behavioral pilot study (N = 102 participants, see data repository) allowed us to match younger and older participants with respect to discrimination performance in the shape condition: age-specific effects were also observed if the baseline performance (feature-alone conditions) was comparable.

Third, the order of conditions was not counterbalanced, and the binding condition was always presented last. Although position effects must be considered in aging studies on working memory span [46], we doubt that proactive interference contributes to age-specific binding effects. Most importantly, the crucial interaction of ‘condition’ and ‘age’ can be observed consistently in the first (F(2,92) = 18.46, *p* < 0.001, η_p_^2^ = 0.286) and second (F(2,92) = 17.39, *p* < 0.001, η_p_^2^ = 0.274) experimental run.

Finally, most of the participants can be related to a WEIRD social background [47] which must be taken into consideration. In addition to the limited sample size, a generalizability of the results requires a larger and more heterogenous sample.

We have no reason to believe that the results depended on other characteristics of the participants, materials, or context.

## 5. Conclusions

The effects of aging in a change detection task requiring the temporary binding of object features cannot be attributed to differences in stimulus encoding. Rather, age-specific ERP effects indicate an additional allocation of resources in order to maintain the internal representation. This additional neural effort in the stage of encoding does not prevent swap errors from being more frequently observed in older participants. The increase in false alarms reflects that older people are more susceptible to recall errors if the cue features in the test array are highly similar to the study array [48]. These age-specific effects are relevant to the recently proposed use of a feature-binding task as a tool in clinical neuropsychology [49].

## Figures and Tables

**Figure 1 brainsci-13-01389-f001:**
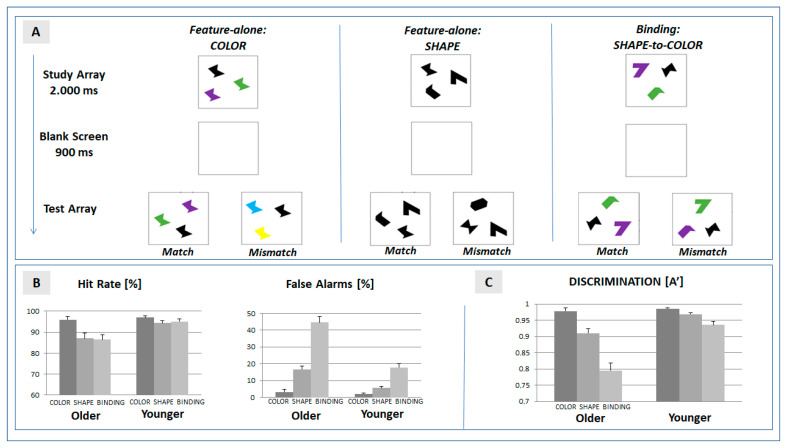
(**A**) Setup of the change detection task. Features of objects presented in the test array were required to match with features presented in the study array. In the feature-alone conditions, color or shape, respectively, were matched. In the binding condition, the shape–color assignment were required to match in the successive arrays; (**B**) mean hit and false alarm rates in the three experimental conditions (color, shape, and binding) separated for the two experimental groups (older and younger participants); and (**C**) mean discrimination ability (A’) in the experimental conditions. Error bars represent the standard error of mean.

**Figure 2 brainsci-13-01389-f002:**
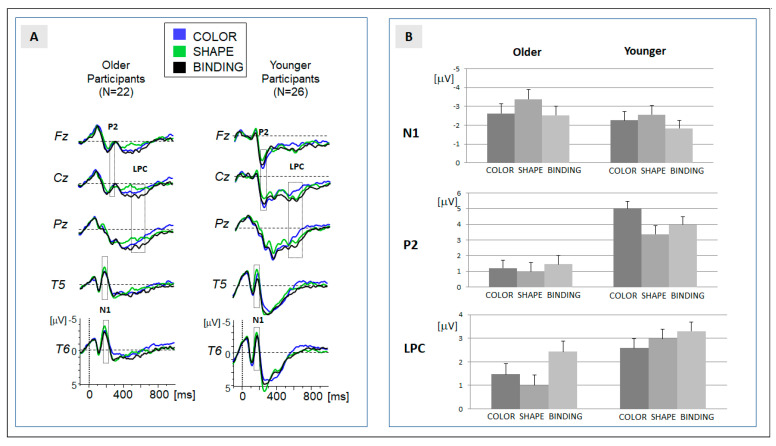
(**A**) Grand-averaged ERPs for the experimental groups ‘older’ (**left**) and ‘younger’ (**right**). Traces for the three conditions are superimposed, and temporal regions of data analysis (N1, P2, and LPC) are indicated; and (**B**) means of the amplitudes for the three ERP components, separated for the experimental group and condition. Error bars represent the standard error of mean.

**Table 1 brainsci-13-01389-t001:** Descriptive statistics. Mean values and confidence intervals for the discrimination ability (A’) and mean amplitudes of the ERP components (N1, P2, LPC). Upper and lower limits of confidence intervals (95%) are presented in brackets.

			OLDER			YOUNGER	
		Color	Shape	Binding	Color	Shape	Binding
A’	M	0.98	0.91	0.8	0.99	0.97	0.94
	CI	[0.96, 0.99]	[0.89, 0.93]	(0.76, 0.83]	[0.97, 1]	[0.95, 0.99]	[0.90, 0.97]
N1	M	−2.63	−3.38	−2.53	−2.26	−2.56	−1.83
	CI	[−3.65, −1.61]	[−4.40, −2.36]	[−3.47, −1.60]	[−3.20, −1.32]	[−3.49, −1.62]	[−2.69, −0.97]
P2	M	1.2	1	1.46	5.01	3.38	3.98
	CI	[0.15, 2.25]	[−0.14, 2.13]	[0.36, 2.55]	[4.04, 5.97]	[2.33, 4.43]	[2.97, 4.98]
LPC	M	1.67	1	2.44	2.6	2.99	3.3
	CI	[0.79, 2.56]	[0.106, 1.89]	[1.58, 3.30]	[1.79, 3.41]	[2.17, 3.81]	[2.51, 4.09]

**Table 2 brainsci-13-01389-t002:** Results of the ANOVA (effects of ‘condition’ and ‘condition x age’) and subsequent post hoc comparisons for the behavioral data (A’) and ERP amplitudes (N1, P2, PLC). (n.i. = not indicated by the superordinated interaction).

Variable/Contrast	Factor Condition	Factor Condition x Age
A’	F(2,92) = 73.49, *p* < 0.001, η_p_^2^ = 0.615	F(2,92) = 24.48, *p* < 0.001, η_p_^2^ = 0.347
Binding vs. Color	F(1,46) = 110.52, *p* < 0.001, η_p_^2^ = 0.706	older: F(1,21) = 72.19, *p* < 0.001, η_p_^2^ = 0.775 younger: F(1,25) = 29.66, *p* < 0.001, η_p_^2^ = 0.543
Binding vs. Shape	F(1,46) = 41.48, *p* < 0.001, η_p_^2^ = 0.474	older: F(1,21) = 24.77, *p* < 0.001, η_p_^2^ = 0.541 younger: F(1,25) = 17.08, *p* < 0.001, η_p_^2^ = 0.406
Color vs. Shape	F(1,46) = 59.17, *p* < 0.001, η_p_^2^ = 0.563	older: F(1,21) = 47.02, *p* < 0.001, η_p_^2^ = 0.691
		younger F(1,25) = 8.68, *p* = 0.007, η_p_^2^ = 0.258
N1	F(2,92) = 8.22, *p* = 0.001, η_p_^2^ = 0.152	F(2,92) = 0.71, *p* = 0.496, η_p_^2^ = 0.015
Binding vs. Color	F(1,46) = 2.15, *p* = 0.150, η_p_^2^ = 0.045	n.i.
Binding vs Shape	F(1,46) = 12.06, *p* = 0.001, η_p_^2^ = 0.208	n.i.
Color vs. Shape	F(1,46) = 8.16, *p* = 0.006, η_p_^2^ = 0.151	n.i.
P2	F(2,92) = 6.43, *p* = 0.002, η_p_^2^ = 0.123	F(2,92) = 4.66, *p* = 0.012, η_p_^2^ = 0.092
Binding vs. Color	F(1,46) = 2.48, *p* = 0.122, η_p_^2^ = 0.051	older: F(1,21) = 0.43, *p* = 0.517, η_p_^2^ = 0.020
		younger: F(1,25) = 10.65, *p* = 0.003, η_p_^2^ = 0.299
Binding vs. Shape	F(1,46) = 3.57, *p* = 0.065, η_p_^2^ = 0.072	older: F(1,21) = 1.71, *p* = 0.205, η_p_^2^ = 0.075
		younger: F(1,25) = 2.02, *p* = 0.167, η_p_^2^ = 0.075
Color vs. Shape	F(1,46) = 14.35, *p* < 0.001, η_p_^2^ = 0.238	older: F(1,21) = 0.47, *p* = 0.500, η_p_^2^ = 0.022
		younger: F(1,25) = 19.85, *p* < 0.001, η_p_^2^ = 0.443
LPC	F(2,92) = 9.25, *p* < 0.001, η_p_^2^ = 0.167	F(2,92) = 4.23, *p* = 0.017, η_p_^2^ = 0.084
Binding vs. Color	F(1,46) = 11.13, *p* = 0.002, η_p_^2^ = 0.195	older: F(1,21) = 5.20, *p* = 0.033, η_p_^2^ = 0.198
		younger: F(1,25) = 5.93, *p* = 0.022, η_p_^2^ = 0.192
Binding vs. Shape	F(1,46) = 14.08, *p* < 0.001, η_p_^2^ = 0.234	older: F(1,21) = 15.16, *p* = 0.001, η_p_^2^ = 419
		younger: F(1,25) = 1.11, *p* = 0.302, η_p_^2^ = 0.043
Color vs. Shape	F(1,46) = 0.49, *p* = 0.487, η_p_^2^ = 0.011	older: F(1,21) = 4.26, *p* = 0.052, η_p_^2^ = 169
		younger: F(1,25) = 2.57, *p* = 0.121, η_p_^2^ = 0.093

## Data Availability

Publicly available datasets were analyzed in this study. This data can be found here: https://osf.io/nw6m5/?view_only=fe52ddb77dd14fa3a9e259c2abc79a10 (accessed on 5 September 2023).

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
