# Peer review of "Age-Specific Effects of Visual Feature Binding"

_brainsci, 2023, doi:10.3390/brainsci13101389_

Round 1

Reviewer 1 Report

Major points

-       The compensatory account of the age-related increase in amplitude of LPC ERP component is not supported by any evidence (e.g., brain-behavior correlations which in older participants showed results far from significance: ps >.4, longitudinal evidence etc.). THerefore the main findings of this study remain without a convicing explanation.

-       Line 111: “Presentation of the subsequent test array was determined by the response 111 of the participant”. In this way, stimulus offset might determine ERP changes that differ across participants and groups, depending on their response speed differences (which unfortunately vary with age).

-       Especially for the early components N1 and P2, it could be interesting to also analyze peak latency and not only mean amplitude.

-       The conclusion section could be improved. For instance, the compensatory explanation is taken for granted and not supported directly in this study. Moreover, towards the the authors write: “These age-specific effects are relevant to the recently proposed use of a feature-binding task as a tool in clinical neuropsychology”: how aer these results relevant in the neuropsychology field? Be clearer)

Minor points

-       Despite the justification through power analysis, the sample sizes for the two age groups (about 22-26 participants each) are quite small and this might potentially represent a threat for replicability. Moreover, related to power issues, the number of trials per condition was quite low in general, but especially for the ERP analysis. More specifically, older adults contributed to ERPs with a smaller number of clean trials per condition on average (about 31-37 vs. about 41-49 for younger adults), which might have decreased the signal-to-noise ratio and increased within group variability for this group.

-       This is a comment on writing style: I would suggest using the more politically correct expressions “younger” and “older” rather than “young” and “old”, when referring to different aging groups

-       Line 34: “is still open see [5].” Insert a comma or parenthesis after “open” and before “see”.

-       Data is the plural form of the latin datum, please use it accordingly (e.g., line 94: “data of nine participants was not” should be changed as “data of nine participants WERE not” etc.).

-       Lines 193-194: “in the majority 193 of the participant.”: “participant” should be changed with “participants”.

-       Line 194: “global filed power”. The authors very likely meant “global field power”.

-       Lines 197-198: data were analyzed running a repeated-measure analysis of variance (ANOVA): given that this analysis included both within-subject and between-subjects factors, it would be better to label it as mixed ANOVA.

A few minor slips are specified in my comments to authors.

Author Response

At first, thank you very much for your comments. They lead to major changes in the discussion of the results.We hope to have met all of your points.

Major points

#1 The compensatory account of the age-related increase in amplitude of LPC ERP component is not supported by any evidence (e.g., brain-behavior correlations which in older participants showed results far from significance: ps >.4, longitudinal evidence etc.). THerefore the main findings of this study remain without a convicing explanation.

Thank you for raising this point. First of all, we are sorry that we didn’t define – and categorize -  the neural process correctly. The task-specific effect is exclusively seen in the LPC component. In the revised manuscript, we describe in more detail which cognitive processes has been related to this component in working memory tasks. We also clarifiy what an age-specific increase on this component might reflect.

Line 408 – 416: The age-specific ERP effect can be linked to a selective neural overactivation which might serve a compensation process [(Smart, Segalowitz, Mulligan, & MacDonald, 2014)]. The centro-parietal LPC has already been observed in working memory task and has been associated with resource allocation (Fortin, Grondin, & Blanchet, 2021; He, Zhang, Li, & Guo, 2015). A more-frontal expression of this component has been related to the maintenance of working memory content (Gao et al., 2011). Correspondingly, the increase of the LPC in the group of old participants might signal that participants recruit more attentional resources in the encoding of the intra-item association and/or that neural networks are more engaged in the attempt to maintain the internal representation of the complex stimuli (Zhou, Li, Broster, Niu, & Wang, 2015)

We also discuss whether this process can be categorized as a compensation process. We follow the reviewer that ‘compensation’ often implies a corresponding effect in cognitive performance. Since the ERP effect and discrimination performance are not correlated, the ERP effect is phrased in a different term.

Line 442 -452: Notably, this process is not directly related to the aforementioned LPC enhancement in older adults. Neither in young nor in old participants, the ERP effect was correlated with discrimination performance. Therefore, it is questionable whether the increase in neural activity reflects an effective neural compensation process. In contrast to previous encoding-related ERP-effects (Paller & Wagner, 2002), the increase of LPC activity observed in older participants is not associated with an increase in discrimination performance. This is also highlighted by the posterior topography of the LPC effect: Aging-related ERP effects which are associated with an increase in the performance in a working memory tasks has been related to frontal processes (Lubitz, Niedeggen, & Feser, 2017; van Dinteren, Arns, Jongsma, & Kessels, 2014). This indicates that the LPC effect observed in the older participants in this study is related to a different neural network not engaged in an effective neural compensation (Cabeza et al., 2018).

This rephrasing lead to changes in the abstract and in the limitations:

Line 23-24 “However, the selective neural overactivation in the encoding phase observed in older participants does not prevent swap errors in the subsequent retrieval phase.”

Line 489 - 490: This the additional neural effort in the stage of encoding does not prevent the swap errors more frequently observed in older participants.

#2 Line 111: “Presentation of the subsequent test array was determined by the response 111 of the participant”. In this way, stimulus offset might determine ERP changes that differ across participants and groups, depending on their response speed differences (which unfortunately vary with age).

This paper is focused on the ERP responses to the study array, and on the encoding processes. The study array was presented for a fixed duration of 2.000 ms , and the ERP effects are not affected by the offset of the display. Moreover, the study array (and the corresponding ERP effects) are not affected by any motor response. We highlight this in the revised version of the manuscript.

Line 187 – 189: ‘Vision analyzer’ (version 2.1, Brain products, Gilching, Germany) was used for offline analysis which was focused on the ERPs triggered in the study array, and which can be linked to the encoding phase:

Line 200 – 202: ERPs triggered by the onset of the test array were also analyzed. As indicated by the grand-averaged ERPs presented in Figure S1, binding-specific effects cannot be observed. This also applies for the time range of the N400, in which binding-specific effects have been reported [(Saiki, 2016)].

#3: Especially for the early components N1 and P2, it could be interesting to also analyze peak latency and not only mean amplitude.

An analysis of the peaks of the N1 and P2 has been conducted. As stated in our manuscript, the data are available in the repository. However, we now refer to the results – confirming the analysis of mean amplitudes – in more detail in the result section.

Line 220-224: Please note that we additionally applied an analysis of peak amplitudes for both components (module ‘peak detection’ in ‘Vision analyzer’ version 2.1, Brain products, Gilching, Germany). Data are available in the data repository, and confirm the results of the analysis of mean amplitudes reported in the following.

Line 322-324: Please note that the result of the peak analysis (see supplement) confirms this result. An effect of N1 latency was not observed. 

Line 332-333: This effect was confirmed by the peak analysis (see supplement). An effect of N1 latency was not observed. 

#4: The conclusion section could be improved. For instance, the compensatory explanation is taken for granted and not supported directly in this study. Moreover, towards the the authors write: “These age-specific effects are relevant to the recently proposed use of a feature-binding task as a tool in clinical neuropsychology”: how aer these results relevant in the neuropsychology field? Be clearer)

The term ’compensation’ has now been re-phrased (see point 1). The impact of the binding task in neuropsychological diagnostics has now been introduced in the introduction in more detail:

Line 73-77: These research questions are also of clinical relevance: As pointed out by Logie and colleagues (2015), the binding task is a sensitive neuropsychological marker in Alzheimer’s Disease (AD). More importantly, the task supports the differentials diagnosis since selective bindings deficits cannot be observed in aging or depression (Parra, Abrahams, Logie, & Della Sala, 2010).

Minor points

#1 Despite the justification through power analysis, the sample sizes for the two age groups (about 22-26 participants each) are quite small and this might potentially represent a threat for replicability. Moreover, related to power issues, the number of trials per condition was quite low in general, but especially for the ERP analysis. More specifically, older adults contributed to ERPs with a smaller number of clean trials per condition on average (about 31-37 vs. about 41-49 for younger adults), which might have decreased the signal-to-noise ratio and increased within group variability for this group.

Thank you for raising this important issue. Unfortunately, we cannot increase the signal-to-noise ratio in the group of the older participants. However, to provide a test of the stability of the effects, we can match the signal-to-noise ratio in the groups. To this end, we reduced the number of EEG segments for each young participant by applying a random selection procedure. The averaged signals relied on a number of trials which was matched for the data of an older participant.

As for the ERP component of interest, the LPC, the descriptive statistics remained stable:

Color: M: 2.58 [CI: 1,75, 3,42] , corresponding data reported in the manuscript: M: 2.6, (CI: 1.79, 3.41)

Form: M: 3.24 [CI: 2,37, 4,10] , corresponding data reported in the manuscript: M: 2,99 (CI: 2.17, 3,81]

Binding: M: 3.44 [CI: 2,49, 4,39] , corresponding data reported in the manuscript: M:: 3,3 (CI: 2,51, 4,09]

Even more importantly, the crucial interaction (COND X GROUP) can be replicated:

F(2,92) = 4,65, p=0.013, eta_square: 0,092

corresponding data reported in the manuscript:: F(2,92) = 4,23, p=0.017, eta_square: 0,084

Although these results (data are available in the repository) indicate a stability of the crucial ERP effect, we agree that results must be replicated. This is now mentioned in the limitations section.

Line 478 – 480: Finally, most of the participants can be related to a WEIRD social background (Henrich, Heine, & Norenzayan, 2010) which has to be considered. In addition to the limited sample size, a generalizability of the results, requires a larger and more heterogenous sample.

#2: This is a comment on writing style: I would suggest using the more politically correct expressions “younger” and “older” rather than “young” and “old”, when referring to different aging groups

Thank you for this advice. We have changed the expressions from “young” to “younger” and from “old” to “older” throughout the manuscript.

#3: Line 34: “is still open see [5].” Insert a comma or parenthesis after “open” and before “see”.

Done

#4: Data is the plural form of the latin datum, please use it accordingly (e.g., line 94: “data of nine participants was not” should be changed as “data of nine participants WERE not” etc.).

Done:

#5: Lines 193-194: “in the majority 193 of the participant.”: “participant” should be changed with “participants”.

Done.

#6: Line 194: “global filed power”. The authors very likely meant “global field power”.

Done.

#7: Lines 197-198: data were analyzed running a repeated-measure analysis of variance (ANOVA): given that this analysis included both within-subject and between-subjects factors, it would be better to label it as mixed ANOVA.

Done.

Reviewer 2 Report

The paper titled "Age-Specific Effects of Visual Feature Binding" addresses the complex issue of how aging affects the binding of visual features, such as color and shape, in working memory. The study utilizes event-related brain potentials (ERPs) to investigate the electrophysiological processes involved in feature binding and how they may change with age. 

The paper has several strengths:

1. Introduction: The paper starts with well-defined research questions about the nature of feature binding processes and whether they are influenced by age. These questions provide a clear focus for the study.

2. Materials and Methods

2.1. Experimental Design: The study employs a carefully designed experimental paradigm that includes both young and older participants, allowing for a direct comparison of age-related effects. The use of a change detection task is appropriate for investigating feature binding.

2.2. Sample Size Justification: The paper provides a rationale for the chosen sample size, which is determined a priori using G*Power based on previous research. This demonstrates a commitment to statistical power.

2.3. Data Analysis: The authors employ rigorous data analysis techniques, including ANOVAs and post-hoc comparisons, to examine both behavioral and ERP data. This approach ensures robust statistical analysis.

3. Results: The paper integrates behavioral data (discrimination ability) with ERP data (N1, P2, LPC) to provide a comprehensive understanding of how feature binding and age interact. This multi-method approach enhances the study's credibility.

4. Discussion: The discussion section effectively interprets the results in the context of the research questions. The authors acknowledge discrepancies with previous findings and propose explanations, providing a nuanced understanding of the results.

However, there are some aspects that could be improved. The paper mentions ERP components (N1, P2, LPC) without providing a detailed explanation of what these components represent in the context of the study. More background information on these components would benefit readers who may not be familiar with ERP research.

Overall, this paper provides valuable insights into the age-specific effects of visual feature binding using a well-designed experimental approach and a combination of behavioral and ERP data.

***

Your reviewer.

Author Response

#1: However, there are some aspects that could be improved. The paper mentions ERP components (N1, P2, LPC) without providing a detailed explanation of what these components represent in the context of the study. More background information on these components would benefit readers who may not be familiar with ERP research.

Thank you for your evaluation. Based on your final remark, we take the opportunity to provide a more-detailed description on the functional characteristics of the components of interest. The categorization is also helpful for the discussion: Here, several cognitive processes attached to the task-specific LPC effect are now mentioned.

 Line 41-52: “First, the posterior N1 component which is generated in the extrastriate cortex and can be modulated by attention (Gazzaley et al., 2008). In complex visual task, the N1 has been assumed to reflect the processing of visual stimulus configuration (Mazza & Caramazza, 2015). Changes in the N1 amplitude has previously been related to the success of a working memory program (Berry et al., 2010). Second,  the fronto-central P2 component which has been related to the perception and processing of salient stimuli, and its expression also depends on the complexity of visual stimuli (Key, Dove, & Maguire, 2005). In working memory tasks, P2 amplitude might serve as an index of attentional resources used in the encoding procedure (Missonier et al., 2004). Finally, a late positive component (LPC) which has been identified in a wide range of cognitive tasks. In memory task, the LPC is assumed to reflect the engagement of elaborated encoding and storage processes (Friedman & Trott, 2000). In working memory, the expression of the LPC is assumed to reflect the recruitment of attentional resources (Fortin et al., 2021) and the maintenance of working memory (He et al., 2015)

Round 2

Reviewer 1 Report

The authors have addressed most of my concerns. I am happy to recommend this paper for publication now.

Please change "old" with "older" and "young" with "younger", when describing the two age groups, as in some instances this change was not implemented (e.g., table 1, figure 2 etc.).

Please change "old" with "older" and "young" with "younger", when describing the two age groups, as in some instances this change was not implemented (e.g., table 1, figure 2 etc.).